# Nintedanib Reduces Neutrophil Chemotaxis via Activating GRK2 in Bleomycin-Induced Pulmonary Fibrosis

**DOI:** 10.3390/ijms21134735

**Published:** 2020-07-02

**Authors:** Wei-Chih Chen, Nien-Jung Chen, Hsin-Pai Chen, Wen-Kuang Yu, Vincent Yi-Fong Su, Hao Chen, Huai-Hsuan Wu, Kuang-Yao Yang

**Affiliations:** 1Department of Chest Medicine, Taipei Veterans General Hospital, Taipei 112, Taiwan; wiji.chen@gmail.com (W.-C.C.); wkyu2@vghtpe.gov.tw (W.-K.Y.); asura811218@gmail.com (H.C.); purplewings0401@gmail.com (H.-H.W.); 2Faculty of Medicine, School of Medicine, National Yang-Ming University, Taipei 112, Taiwan; hpchen5@vghtpe.gov.tw (H.-P.C.); bsbipoke@hotmail.com (V.Y.-F.S.); 3Institute of Emergency and Critical Care Medicine, School of Medicine, National Yang-Ming University, Taipei 112, Taiwan; 4Institute of Microbiology and Immunology, School of Life Sciences, National Yang-Ming University, Taipei 112, Taiwan; nienjung.chen@gmail.com; 5Cancer Progression Research Center, National Yang-Ming University, Taipei 112, Taiwan; 6Division of Infectious Diseases, Department of Medicine, Taipei Veterans General Hospital, Taipei 112, Taiwan; 7Institute of Physiology, School of Medicine, National Yang-Ming University, Taipei 112, Taiwan; 8Department of Internal Medicine, Taipei City Hospital, Taipei 112, Taiwan

**Keywords:** nintedanib, neutrophil, chemokine (C-X-C motif) receptor 2 (CXCR2), G protein-coupled receptor kinase 2 (GRK2), pulmonary fibrosis

## Abstract

Neutrophils are involved in the alveolitis of idiopathic pulmonary fibrosis (IPF). However, their pathogenic mechanisms are still poorly understood. Nintedanib has antifibrotic and anti-inflammatory activity in IPF. This study aimed to investigate the regulatory mechanism of nintedanib on neutrophil chemotaxis in bleomycin (BLM)-induced pulmonary fibrosis. Nintedanib was administered via oral gavage to male C57BL/6 mice 24 h after a bleomycin intratracheal injection (1.5 U/kg). Lung histopathological findings, the expression of cytokines, and the regulatory signaling pathways of neutrophil chemotaxis were analyzed. The effect of nintedanib was also investigated in a mouse model with adoptive neutrophil transfer in vivo. Nintedanib significantly decreased the histopathological changes and neutrophil recruitment in BLM-induced pulmonary fibrosis. Nintedanib mediated a downregulation of chemokine (C-X-C motif) receptor 2 (CXCR2) and very late antigen 4 (VLA-4) expression, as well as an upregulation of G protein-coupled receptor kinase 2 (GRK2) activity in peripheral blood neutrophils in BLM-induced pulmonary fibrosis. Nintedanib also decreased the activation of endothelial cells by the decreased expression of vascular cell adhesion molecule 1 (VCAM-1). The effect of nintedanib on regulating neutrophil chemotaxis was also confirmed by a mouse model with adoptive neutrophil transfer in vivo. In conclusion, nintedanib reduces neutrophil chemotaxis and endothelial cell activation to regulate the severity of BLM-induced pulmonary fibrosis. These effects are associated with an enhancement of GRK2 activity and a reduction in CXCR2 and VLA-4 expression on neutrophils and a decrease in VCAM-1 expression on endothelial cells.

## 1. Introduction

In end-stage interstitial lung disease (ILD) and in the fibrotic phase of acute respiratory distress syndrome (ARDS), pulmonary fibrosis is commonly observed [1,2]. Histologically, the hallmarks of pulmonary fibrosis include the excessive deposition of a disorganized matrix and collagen, the proliferation of mesenchymal cells, the destruction of normal lung structure, and the formation of honeycomb cysts [3]. Patients affected by idiopathic pulmonary fibrosis (IPF), the most common form of ILD, have a median survival of only 27.4 to 55.6 months [4]. The poor survival of IPF reflects the insufficient knowledge of its pathogenesis and the absence of effective treatments to slow or even reverse the disease process [3].

Bleomycin-induced pulmonary fibrosis has been used for years as a model of fibrotic lung diseases in animal studies. It involves acute inflammation in the alveolar epithelium followed by fibrosis, which is also observed in IPF and ARDS [5,6]. Our previous study demonstrated that the early administration of induced pluripotent stem cells might reduce the levels of cytokines and chemokines that mediate inflammation and fibrosis in murine models of bleomycin-induced pulmonary fibrosis [7]. Therefore, developing an effective treatment to alleviate lung inflammation during the early stage of fibrosis is important.

Both innate and adaptive immunity contribute to fibrogenesis at several cellular and noncellular levels in pulmonary fibrosis. Immune cells, including neutrophils, macrophages, T cells, and fibrocytes, as well as soluble mediators, such as cytokines and chemokines, are all involved in the pathogenesis of pulmonary fibrosis [8]. Among them, neutrophils can modify their tissue environment by releasing proteases, oxidants, cytokines, and chemokines in pulmonary fibrosis [9]. Elevated levels of neutrophil elastase, as evidence of neutrophil activation, were demonstrated in the lung parenchyma and in both the bronchoalveolar lavage (BAL) fluid and sera of patients with IPF [10]. The inhibition of neutrophil elastase has been shown to reduce the process of pulmonary fibrosis [11,12].

Very late antigen 4 (VLA-4) has been recognized on activated neutrophils and contributes to the adhesion, recruitment, rolling, and migration of neutrophils [13,14,15]. In endotoxin-induced lung inflammation, VLA-4 mediates neutrophil migration to the lung and is involved in pulmonary vascular and epithelial permeability [16]. Vascular cell adhesion molecule 1 (VCAM-1) is an endothelial ligand for VLA-4 [17]. The interaction between VCAM-1 and VLA-4 produces mechanical strength, which allows the complex to resist shear forces imposed on it by the bloodstream [18]. It also plays a role in the transendothelial migration of leukocytes [19,20]. Bleomycin promotes VCAM-1 expression in pulmonary microvascular endothelial cells and neutrophil adhesion [21].

Nintedanib is a small molecule inhibitor of tyrosine kinase receptors, including vascular endothelial growth factor receptor, fibroblast growth factor receptor, and platelet-derived growth factor receptor [22]. In clinical trials, nintedanib led to a significant reduction in the rate of decline in pulmonary function in patients with IPF, reduced the time to the first acute exacerbation, and slowed disease progression with acceptable tolerability and safety [23,24].

In addition, nintedanib had both antifibrotic and anti-inflammatory activity in an animal model of pulmonary fibrosis. Neutrophils in BAL fluid and proinflammatory cytokines were reduced after either preventive or therapeutic treatment with nintedanib in pulmonary fibrosis [25]. However, the detailed mechanism is unknown. The goal of this study was to investigate the role of nintedanib in regulating neutrophils in bleomycin-induced pulmonary fibrosis.

## 2. Results

### 2.1. Effects of Nintedanib on the Histopathology of Bleomycin-Induced Pulmonary Fibrosis

Lung inflammation and fibrosis induced by the intratracheal injection of bleomycin were evidenced by increased neutrophil infiltration in lung tissue, progressive diffuse alveolar fibrosis, focally dense fibrosis, and epithelial hyperplasia in alveolar ducts with hematoxylin and eosin (HE) staining (Figure 1A). To investigate the role of nintedanib in bleomycin-induced pulmonary fibrosis, we compared lung sections from pulmonary fibrosis mice that underwent treatment with nintedanib via oral gavage and those from mice with pulmonary fibrosis that did not undergo treatment. The histological evaluation of the lungs seven days after bleomycin-induced pulmonary fibrosis showed that lung inflammation and fibrosis were significantly reduced by nintedanib treatment (Figure 1A). Mice treated with nintedanib also had significantly decreased lung injury scores (Figure 1A). Masson’s trichrome staining of lung tissue and the Ashcroft score showed progressive pulmonary fibrosis seven days after bleomycin injection (Figure 1B). In contrast, treatment with nintedanib significantly reduced bleomycin-induced pulmonary fibrosis both according to Masson’s trichrome staining and the Ashcroft score (Figure 1B).

### 2.2. Nintedanib Regulates the Expression of Alpha Smooth Muscle Actin (α-SMA) and Collagen-1

Immunohistochemistry (IHC) staining of lung tissue was performed to detect changes in various proteins in lung tissue following the administration of bleomycin. Increases in α-SMA and collagen-1 were observed seven days after bleomycin injection. Moreover, nintedanib treatment significantly reduced α-SMA and collagen-1 levels after bleomycin injection (Figure 1C,D).

### 2.3. Effects of Nintedanib on Proinflammatory Cytokines in Lung Tissues

An ELISA of the whole lung extracts showed that interleukin-1 beta (IL-1β) and macrophage inflammatory protein-2 (MIP-2) levels were significantly higher in the pulmonary fibrosis model group than in the control group seven days after bleomycin injection (Figure 2A,B). For both IL-1β and MIP-2, their levels were significantly reduced following the administration of nintedanib compared to the bleomycin group (Figure 2A,B).

### 2.4. Effects of Nintedanib on Neutrophil Accumulation in the Lung

The intratracheal instillation of bleomycin resulted in a significant increase in the accumulation of neutrophils in the lungs, as demonstrated visually by the lymphocyte antigen 6G (Ly6G) and chemokine (C-X-C motif) receptor 2 (CXCR2) immunofluorescence (IF) staining of the lung (Figure 3A). In contrast, nintedanib treatment was associated with a significant decrease in neutrophil accumulation (Figure 3A).

### 2.5. Effects of Nintedanib on CXCR2 and G Protein-Coupled Receptor Kinase 2 (GRK2) Expression Levels in Peripheral Blood

Flow cytometry and IF staining of peripheral blood were performed to detect Ly6G and CXCR2 on circulating neutrophils. The expression levels of Ly6G and CXCR2 in mouse neutrophils were significantly increased seven days after bleomycin injection. Treatment with nintedanib reduced the expression levels of Ly6G and CXCR2 by mouse neutrophils seven days after bleomycin injection (Figure 3B). IF staining revealed that GRK2 expression by mouse neutrophils in peripheral blood was significantly upregulated in mice treated with nintedanib after bleomycin injection compared with the bleomycin group (Figure 3C).

### 2.6. Nintedanib Prevents Neutrophil Chemotaxis in Mice with Bleomycin-Induced Pulmonary Fibrosis

IHC staining and western blots showed significantly higher expression of VLA-4, a neutrophil integrin and a marker of neutrophil adhesion, seven days after bleomycin injection. Nintedanib treatment significantly reduced the increase in VLA-4 after bleomycin injection (Figure 4A and Figure 5). In addition, IHC staining and western blots showed significantly higher expression of VCAM-1, an integrin receptor on vascular endothelial cells that binds to VLA-4 on neutrophil membranes, seven days after bleomycin injection. Nintedanib treatment decreased the expression of VCAM-1 after bleomycin injection (Figure 4B and Figure 5).

### 2.7. Effects of Nintedanib on Neutrophil Chemotaxis and Migration in Adoptive Transfer In Vivo

Lung inflammation and fibrosis induced by the in vivo adoptive transfer of neutrophils derived from mice intratracheally injected with bleomycin were evident seven days after administration. In contrast, nintedanib treatment significantly attenuated the lung inflammation and fibrosis, as evidenced by both HE staining and the lung injury score (Figure 6A). Masson’s trichrome staining of lung tissue and the Ashcroft score showed progressive pulmonary fibrosis seven days after the in vivo adoptive transfer of neutrophils derived from mice intratracheally injected with bleomycin (Figure 6B). Treatment with nintedanib also significantly reduced pulmonary fibrosis according to both Masson’s trichrome staining and the Ashcroft score (Figure 6B).

## 3. Discussion

To the best of our knowledge, this is the first study reporting the role of nintedanib in the regulation of neutrophil chemotaxis in pulmonary fibrosis. The therapeutic effect of nintedanib occurred through an enhancement of GRK2 activity and a reduction in CXCR2 expression by neutrophils.

Neutrophils were found to be increased in BAL fluid from patients with IPF, and baseline neutrophilia was associated with early mortality [26]. Aside from serine proteases, neutrophils are an important source of matrix metallopeptidases (MMPs), such as MMP-2, MMP-8, and MMP-9, which are involved in pulmonary fibrosis [27,28]. In the present study, we further proved the role of neutrophils in the pathogenesis of bleomycin-induced pulmonary fibrosis. We also found that nintedanib not only had antifibrotic activity, but also exerted an anti-inflammatory function against neutrophils. Interestingly, nintedanib had different extents of inhibitory effect on the inhibition of IL-1β and MIP-2 in our study. As well as neutrophils, IL-1β was also produced by activated macrophages, dendritic cells, and epithelial cells. It was shown to contribute to the progression of pulmonary fibrosis [29]. MIP-2 could be produced by macrophages, monocytes, and epithelial cells. In bleomycin-induced pulmonary fibrosis, significantly increased MIP-2 was correlated to a greater angiogenic response and total lung hydroxyproline content [30,31]. In humans, interleukin-8 (IL-8) could attract neutrophils through their receptors, such as CXCR2. However, mice lack the IL-8 gene [32]. MIP-2 is known as a functional homologue of IL-8 in mice and contributes to a number of neutrophil-dependent diseases in mice models [31].

The preventive administration of nintedanib has been shown to reduce neutrophils and lymphocytes, but not macrophages, in the BAL fluid in silica-induced pulmonary fibrosis. Additionally, the therapeutic administration of nintedanib, starting at day 10, reduced neutrophils and lymphocytes in the BAL fluids in a comparable manner to preventive treatment [25]. Our study further found that early treatment with nintedanib 24 h after bleomycin-induced pulmonary fibrosis could effectively prevent neutrophil infiltration in the lung. Moreover, our study demonstrated that nintedanib downregulated Ly6G and CXCR2 expression levels in circulating neutrophils while upregulating GRK2 expression in pulmonary fibrosis. Evidence suggests that CXCR2 mediates pulmonary fibrosis after bleomycin instillation in mice. The blockade of CXCR2 was associated with inhibition of neutrophil migration into lung parenchyma, the inhibition of fibrogenic cytokines, and decreased pulmonary angiogenesis and fibrosis [33]. In addition, the downregulation of GRK2 was associated with the augmentation of CXCR2-induced neutrophil migration after lipopolysaccharide administration [34]. In our previous study, induced pluripotent stem cells could downregulate neutrophil chemotaxis in endotoxin-induced acute lung injury, an effect associated with the enhancement of GRK2 activity and the reduction of CXCR2 expression [35]. The current study also showed that nintedanib attenuated neutrophil chemotaxis through the upregulation of GRK2 expression and the subsequent downregulation of CXCR2 levels in neutrophils. Besides, nintedanib decreased both VLA-4 expression on neutrophils and VCAM-1 expression on pulmonary endothelial cells. As a result, the recruitment of neutrophils was reduced in bleomycin-induced pulmonary fibrosis.

In patients with IPF, anti-inflammatory therapy, such as corticosteroids, had been extensively used by pulmonologists in an earlier survey [36]. However, in a large clinical trial, combination treatment with prednisolone, *N*-actetylcysteine, and azathioprine in IPF patients not only increased risks of hospitalization, but also mortality compared with the placebo group [37]. In contrast, corticosteroids might be considered in the acute exacerbation of IPF [38]. This reflects the different pathophysiologic processes in different stages of IPF. In fibrosing lung diseases, especially IPF, nintedanib exerts both anti-inflammatory and antifibrotic activity [39]. 

Our study had several limitations. First, the pathogenesis of pulmonary fibrosis involves various types of inflammatory cells with different mechanisms. Although we explored the role of nintedanib on neutrophil chemotaxis, other therapeutic effects of nintedanib on inflammation and fibrosis from bleomycin were not presented. Secondly, we administered nintedanib 24 h after bleomycin injection, and the administration of nintedanib successfully attenuated the lung inflammation, fibrosis, and neutrophil chemotaxis that occurred later. Whether different time points, dosages, and dosing frequencies of nintedanib administration will have similar results requires further investigation. Thirdly, although we performed in vivo experiments and an adoptive transfer study, we did not demonstrate the direct effect of nintedanib on isolated neutrophils to assess their chemotaxis in response to stimuli.

## 4. Materials and Methods 

### 4.1. Experimental Animals

Male C57BL/6 mice (8–12 weeks old with weight 20–25 g) were purchased from the National Experimental Animal Center (Taipei, Taiwan) and kept in standard plastic animal cages with husk bedding at 25 ± 2 °C with a 12-h light/dark cycle. All of the mice were maintained under specific pathogen-free conditions and were provided food and water ad libitum. All experiments were conducted in accordance with the Institutional Animal Care and Use Committee-approved protocols (TVGH IACUC No.2017-076, 06/30/2017).

### 4.2. Animal Model

After anesthesia, the mice received an intratracheal injection of bleomycin sulfate (BLM) (Merck, Darmstadt, Germany) at a dose of 1.5 U/kg in 50 µL phosphate-buffered saline (PBS) to induce pulmonary fibrosis in a protocol adapted from our previous experiments [7]. In addition, the control mice received an intratracheal injection of 50 µL PBS. In designated experiments, mice received either nintedanib (suspended in 300 µL 0.5% hydroxyethyl cellulose (HEC) and administered at a dose of 50 mg/kg daily for five days every week), as modified from a previous study [25] (referred to as w/ Nin-treated BLM mice), or HEC 300 µL (referred to as w/o Nin-treated BLM mice) via oral gavage 24 h after the induction of lung fibrosis. The animals were sacrificed 7, 14, or 21 days post-BLM injection because these time points represent the phases of maximal inflammation (7 days) and fibrosis (14 and 21 days).

### 4.3. Histology and Immunohistochemistry

Lungs from both the nintedanib-treated group and the control group were excised 7, 14, and 21 days after bleomycin-induced pulmonary fibrosis. Then, lung sections were fixed with 4% paraformaldehyde for 10 min, embedded in paraffin and cut into 3 mm thick sections. HE and IHC staining was performed on 4 µm paraffin sections of formalin-fixed lung samples. Staining for α-SMA (1:100, 14395-1-AP, Proteintech, Rosemont, IL, USA), collagen-1 (Abcam, Cambridge, UK), VLA-4 (Cell Signaling TECHNOLOGY, Danvers, MA, USA), and VCAM-1 (Cell Signaling TECHNOLOGY, Danvers, MA, USA) were carried out using Envision^®^ + Dual Link System-HRP (DAB+) kits (K4065, DAKO, Carpinteria, CA, USA). Briefly, the sections were deparaffinized with xylene, dehydrated with ethanol, and subsequently heated in 0.01 M citrate buffer (pH 6.0). Endogenous peroxidase activity was inactivated in 3% H_2_O_2_ for 10 min at room temperature, and the sections were blocked with a blocking buffer (K4065 kit). Samples were then incubated with antibodies (anti-α-SMA, anti-collagen-1, anti-VLA-4, and anti-VCAM-1) at room temperature for 1 h. Secondary anti-mouse and antibody-coated polymer peroxidase complexes (K4065 kit) were then applied for 30 min at room temperature, followed by substrate/chromogen treatment (K4065 kit) and a further incubation for 5 to 15 s at room temperature. Slides were counterstained with hematoxylin (109249, Merck, Darmstadt, Germany) for 10 s and subsequently washed in running water for 10 min. The sections were observed and photographed with a Leica microscope (Leica Camera AG, Wetzlar, Germany) and a SPOT RT^TM^ camera (Diagnostic Instruments, Inc., Sterling Heights, MI, USA). To quantitatively analyze the IHC intensity of the scratched area, the percentage of IHC signal per photographed field was analyzed by image processing software (Image-Pro Plus, Media Cybernetics, Inc., Silver Spring, MD, USA).

### 4.4. Lung Injury Score

To quantify the severity of lung fibrosis by histology, the lung injury score was assessed. Two investigators independently evaluated each HE-stained slide in a blinded manner. To generate the lung injury score, 300 alveoli were counted on each slide at 400× magnification. Within each field, points were assigned according to the predetermined criteria used in a previous study [40]: lung injury score = ([alveolar hemorrhage points/number of fields] + 2 × [alveolar infiltrate points/number of fields] + 3 × [fibrin points/number of fields] + [alveolar septal congestion/number of fields])/total number of alveoli counted. Five mice were used in each group.

### 4.5. Masson’s Trichrome Staining

Lung sections were fixed in 4% paraformaldehyde for 10 min, embedded in paraffin, and cut into 3 µm thick sections. These sections were stained using the Trichrome Stain Kit (#ab150686, Abcam, Cambridge, UK) according to the manufacturer’s instructions.

### 4.6. Ashcroft Scale

Two investigators independently evaluated each Masson’s trichrome-stained slide in a blinded manner. Within each field, points were assigned according to the predetermined criteria used in a previous study [41].

### 4.7. Enzyme-Linked Immunosorbent Assay (ELISA)

Homogenized lung tissue in a lysis buffer consisting of radioimmunoprecipitation assay (RIPA) buffer (475 µL), reagent cocktail (5 µL), and 0.1 M Na_3_VO_4_ 20 µL was centrifuged at 20,000 rpm for 10 min at 4 °C and stored at −20 °C until use. IL-1β (MLB00C, R&D Systems, Inc., Minneapolis, MN, USA) and MIP-2 (MM200, R&D Systems, Inc., Minneapolis, MN, USA) levels in lung tissue were analyzed using Quantikine ELISA kits (M6000B, R&D Systems, Inc., Minneapolis, MN, USA). Lung samples were diluted with calibrator diluent (1:4).

### 4.8. Western Blotting

Homogenized lung tissue in lysis buffer (475 µL RIPA, 5 µL reagent cocktail, and 20 µL 0.1 M Na_3_VO_4_) was centrifuged at 20,000 rpm for 10 min at 4 °C and stored at −20 °C until use. Equal amounts of protein homogenate were resolved on 7.5%–10% sodium dodecyl sulfate–polyacrylamide gel electrophoresis gels and transferred onto polyvinylidene fluoride membranes. Blots were blocked in Tris-buffered saline and polysorbate 20 (TBST, with 1% Tween 20) containing 5% milk and probed with primary antibodies directed toward VCAM-1 (#14694, 1:1000, Cell Signaling TECHNOLOGY, Danvers, MA, USA), VLA-4 (Cell Signaling TECHNOLOGY, Danvers, MA, USA), and β-actin (20536-1-A, 1:5000, Proteintech, Rosemont, IL, USA). Blots were subsequently washed in TBST, incubated with horseradish peroxidase secondary antibodies (1:1000, goat anti-rabbit IgG H+L, ab205718, Abcam, Cambridge, UK), and detected using enhanced chemiluminescence (Pierce Biochemicals, 32106, Thermo Fisher Scientific, Waltham, MA, USA). Each blot was exposed to a film, and densitometry of immunoreactive bands was performed using ImageJ software (National Institutes of Health, Bethesda, MD, USA).

### 4.9. Immunofluorescence

The sections were deparaffinized with xylene, dehydrated with ethanol, and then heated in 0.01 M citrate buffer (pH 6.0). After blocking with 3% fetal bovine serum (FBS) (in PBS) for 60 min at room temperature, antibodies against Ly6G (1:100, LS-C139872, LSBio, Washington, USA) and CXCR2 (1:100, ab14935, Abcam, Cambridge, UK) were applied overnight at 4 °C. The next day, goat anti-rabbit immunoglobulin G (IgG) H+L (Alexa Fluor^®^ 488) (1:400, ab150077, Abcam, Cambridge, UK) and goat anti-rabbit IgG H+L (Cy5^®^) (1:400, ab6564, Abcam, Cambridge, UK) secondary antibodies were incubated at 37 °C for 2 h. The slides were mounted using mounting medium with 4′,6-diamidino-2-phenylindole (DAPI) (H-1200, Vector Laboratories, Burlingame, CA, USA) to obtain nuclear staining. Images of the cells were taken on a FLUOVIEW confocal microscope (FV10i, Olympus America, Melville, NY, USA).

### 4.10. Flow Cytometry

To examine surface CXCR2 expression, mouse whole blood cells with various treatments were stained with equal fluorescein isothiocyanate (FITC)-labeled anti-Ly6G and Alexa647-labeled anti-CXCR2 antibodies (0.125 µg per 10^6^ cells, BioLegend, San Diego, CA, USA) and subjected to flow cytometric analysis with a FACS Canto II flow cytometer (BD Biosciences, San Jose, CA, USA). To examine intracellular GRK2 expression, cells were first stained with appropriate fluorescently labeled antibodies to detect cell surface markers. Cells were fixed, permeabilized with Tween 20, and stained with antibodies against GRK2 (1:50, ab32558, Abcam, Cambridge, UK) and the corresponding phycoerythrin (PE)-labeled secondary antibodies. Stained cells were washed and analyzed in a FACS Canto II flow cytometer. Data were analyzed by FlowJo software (Tree Star, Ashland, OR, USA) with 10,000 events per sample.

### 4.11. Confocal Microscopy

Confocal microscopy was performed to detect intracellular GRK2 expression. Mouse blood cells were subjected to cytospinning and were fixed, permeabilized, and stained with an anti-GRK2 antibody (1:100, ab32558, Abcam, Cambridge, UK) as the primary antibody. Goat anti-rabbit IgG H+L Cy5^®^ (1:400, ab6564, Abcam, Cambridge, UK) was used as the secondary antibody. Stained cells were counterstained with DAPI and analyzed with a FLUOVIEW confocal microscope (FV10i, Olympus America, Melville, NY, USA).

### 4.12. Neutrophil Adoptive Transfer

Neutrophils were isolated from mice after bleomycin treatment for 7 days with magnetic-activated cell sorting systems following the anti-Ly6G microbead kit and protocol (130-092-332, Miltenyi Biotec, Bergisch Gladbach, Germany). Each mouse was seeded with 5 × 10^5^ neutrophils via tail vein injections, and the lungs were collected after 7 days and 14 days.

### 4.13. Statistical Analysis

The mice were prepared and studied concomitantly. Separate mice were used for IHC, lung injury score, Ashcroft scale, ELISA, immunofluorescence, and western blotting analyses. Data are presented as the mean ± standard error of the mean or standard deviation for each experimental group. One-way analysis of variance and the Tukey–Kramer multiple comparisons test or a pairwise Student’s *t*-test were used. A *p* value less than 0.05 was considered significant.

## 5. Conclusions

In conclusion, nintedanib reduces neutrophil chemotaxis to decrease the severity of bleomycin-induced pulmonary inflammation and fibrosis. These effects are associated with an enhancement of GRK2 activity and a reduction in CXCR2 and VLA-4 expression from neutrophils. On the other hand, nintedanib diminishes endothelial cell activation with a decreased VCAM-1 expression.

## Figures and Tables

**Figure 1 ijms-21-04735-f001:**
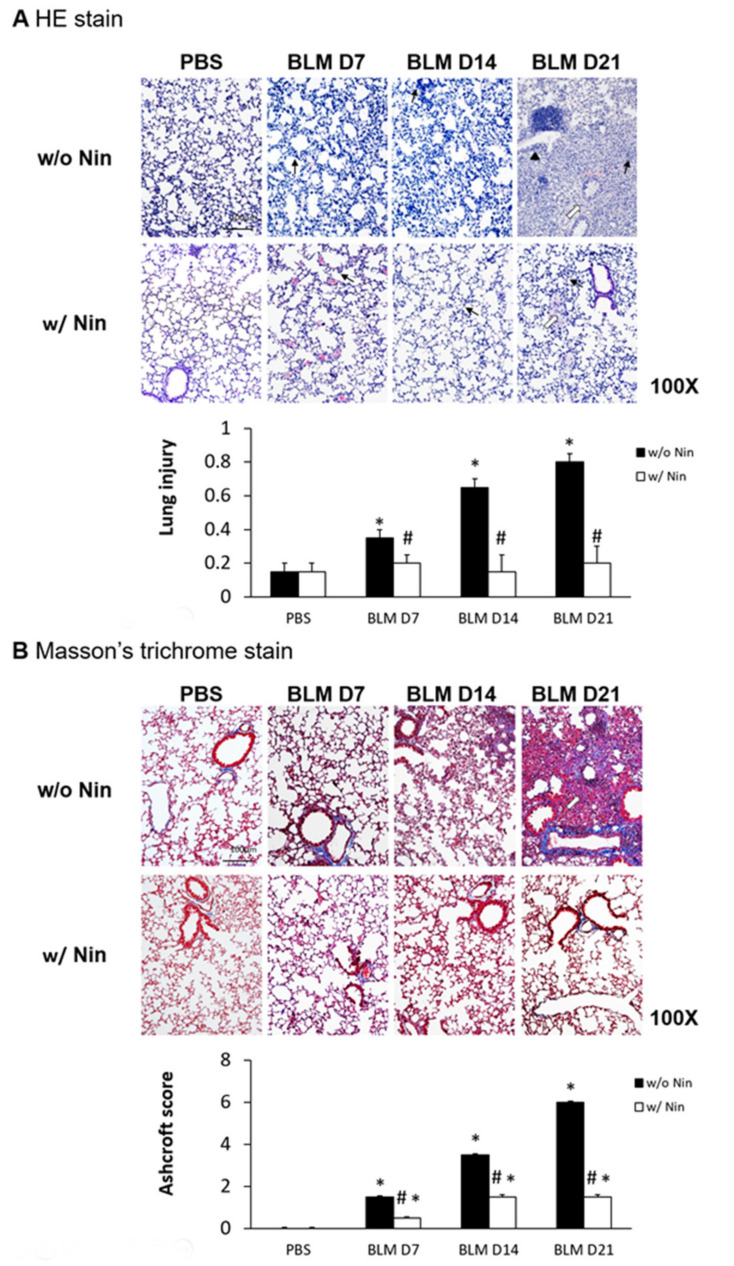
Administration of nintedanib improved histological features and reduced pulmonary inflammation and fibrosis in mice with bleomycin-induced pulmonary fibrosis. (**A**) Hematoxylin and eosin (HE) staining and lung injury scores demonstrated that lung inflammation (arrows: neutrophil infiltration in lung tissue) and fibrosis (open arrows: alveolar fibrosis; arrowhead: epithelial hyperplasia in alveolar ducts) were attenuated in mice receiving nintedanib (Nin) compared with those in the bleomycin (BLM) group seven days after bleomycin injection. (**B**) Pulmonary fibrosis mice that received nintedanib had significantly less pulmonary fibrosis (open arrow: alveolar fibrosis) than mice in the bleomycin group according to Masson’s trichrome staining and the Ashcroft score. (**C**) Immunohistochemical staining demonstrated that the levels of alpha smooth muscle actin (α-SMA) were significantly increased seven days after bleomycin injection. The administration of nintedanib reduced α-SMA levels fourteen days after bleomycin injection. (**D**) Similarly, collagen-1 levels significantly increased with bleomycin-induced pulmonary fibrosis. The administration of nintedanib reduced collagen-1 expression fourteen days after bleomycin injection. * *p* < 0.05 compared to the PBS group; # *p* < 0.05 compared to the BLM w/o Nin group, *n* = 5 per group. PBS: phosphate-buffered saline, w/o: without, w/: with.

**Figure 2 ijms-21-04735-f002:**
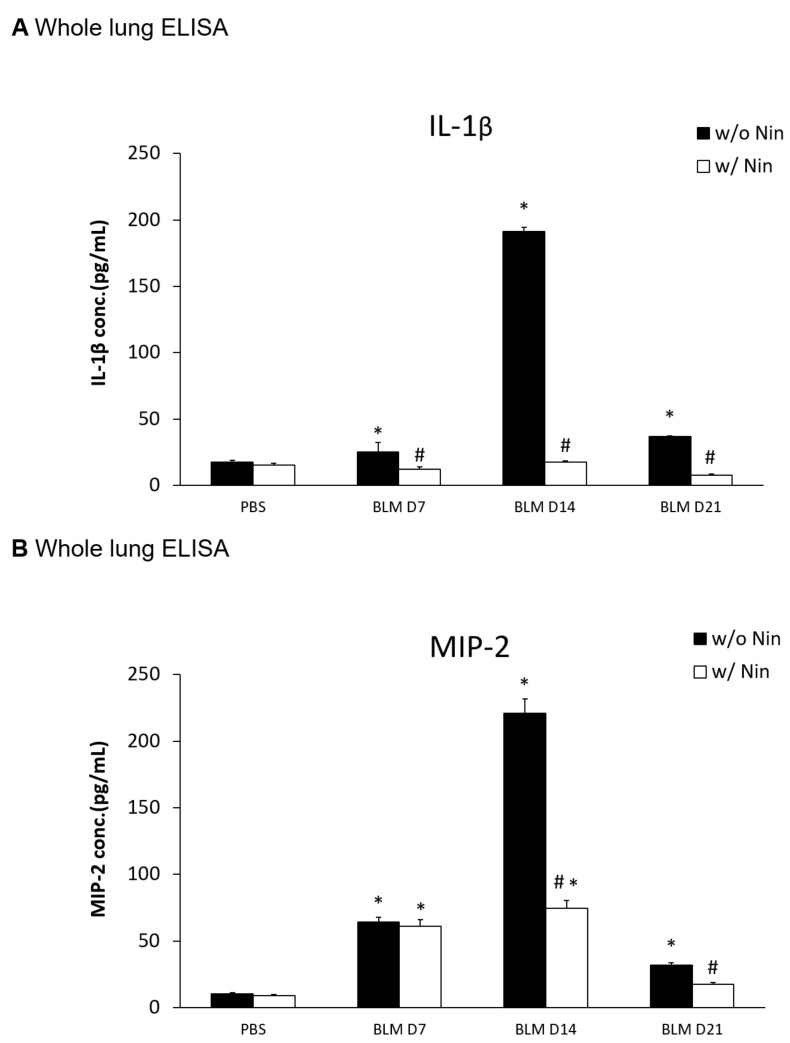
Nintedanib regulates proinflammatory cytokines in mice with bleomycin-induced pulmonary fibrosis. (**A**) Enzyme-linked immunosorbent assay (ELISA) of whole lung extracts revealed that interleukin-1 beta (IL-1β) levels significantly increased seven days after bleomycin (BLM) injection. The administration of nintedanib (Nin) reduced IL-1β levels seven days after bleomycin injection. (**B**) ELISA of the whole lung extracts revealed that macrophage inflammatory protein-2 (MIP-2) levels significantly increased seven days after bleomycin injection. The administration of nintedanib reduced the increase in MIP-2 fourteen days after bleomycin injection. * *p* < 0.05 compared to the PBS group; # *p* < 0.05 compared to the BLM w/o Nin group, *n* = 5 per group. PBS: phosphate-buffered saline, w/o: without, w/: with.

**Figure 3 ijms-21-04735-f003:**
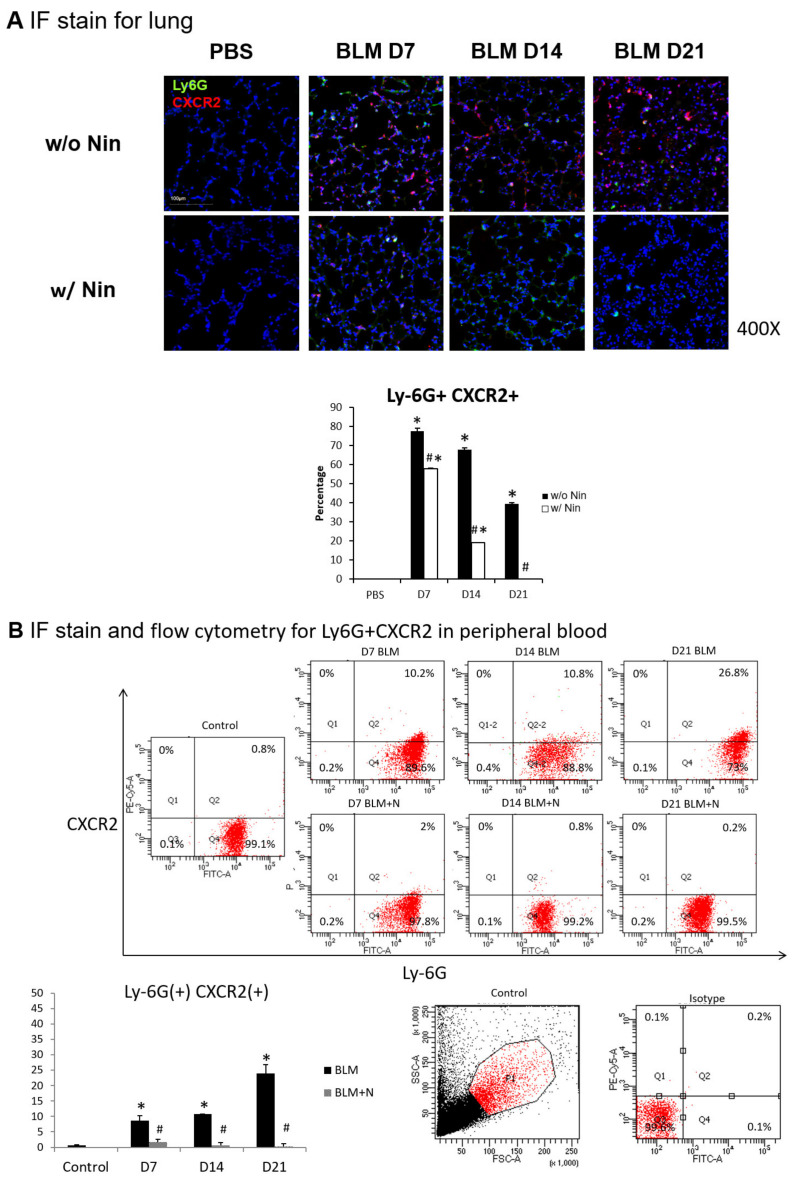
Administration of nintedanib prevented neutrophil accumulation in lung tissue and reduced the number of neutrophils in the peripheral blood of mice with bleomycin-induced pulmonary fibrosis. (**A**) Neutrophil accumulation in the lung was significantly increased seven days after bleomycin (BLM) injection, as seen by immunofluorescence (IF) double staining for lymphocyte antigen 6G (Ly6G) and chemokine (C-X-C motif) receptor 2 (CXCR2). The administration of nintedanib (Nin) significantly reduced neutrophil accumulation in the alveoli seven days after bleomycin injection. (**B**) The expression levels of Ly6G and CXCR2 in mouse neutrophils in peripheral blood were significantly increased seven days after bleomycin injection, shown by IF staining and flow cytometry. The administration of nintedanib significantly reduced the expression levels of Ly6G and CXCR2 seven days after bleomycin injection. (**C**) G protein-coupled receptor kinase 2 (GRK2) expression by mouse neutrophils in peripheral blood was significantly upregulated in mice treated with nintedanib fourteen days after bleomycin injection compared with that in the bleomycin group. * *p* < 0.05 compared to the PBS group; ^#^
*p* < 0.05 compared to the BLM w/o Nin group, *n* = 5 per group. PBS: phosphate-buffered saline, w/o: without, w/: with.

**Figure 4 ijms-21-04735-f004:**
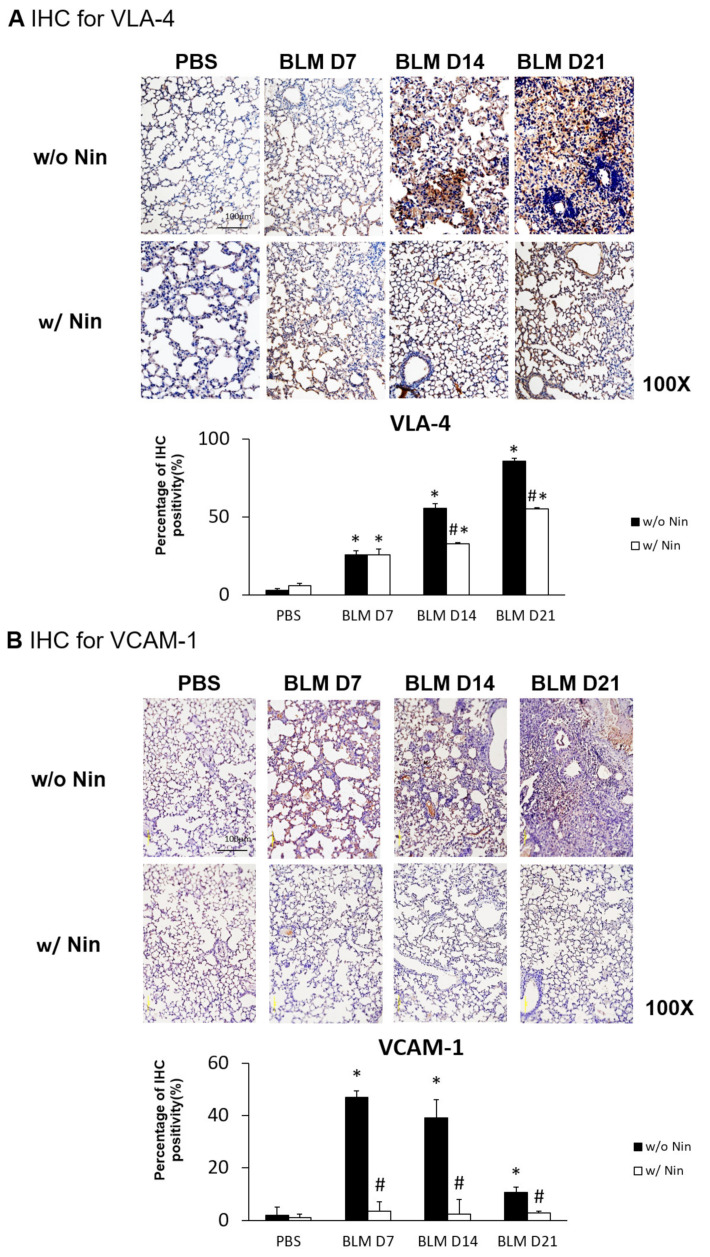
Nintedanib prevents neutrophil chemotaxis in mice with bleomycin-induced pulmonary fibrosis. (**A**) The expression of very late antigen 4 (VLA-4), a neutrophil integrin, was significantly higher seven days after bleomycin injection, shown by immunohistochemical (IHC) staining. Nintedanib (Nin) treatment significantly reduced the increase in VLA-4 fourteen days after bleomycin injection. (**B**) Vascular cell adhesion molecule 1 (VCAM-1), an integrin receptor on vascular endothelial cells, binds to VLA-4 on neutrophil membranes, and it had significantly higher expression seven days after bleomycin injection than before bleomycin injection, shown by IHC staining. Nintedanib treatment decreased the expression of VCAM-1 seven days after bleomycin injection. * *p* < 0.05 compared to the PBS group; # *p* < 0.05 compared to the BLM w/o Nin group, *n* = 5 per group. PBS: phosphate-buffered saline, w/o: without, w/: with.

**Figure 5 ijms-21-04735-f005:**
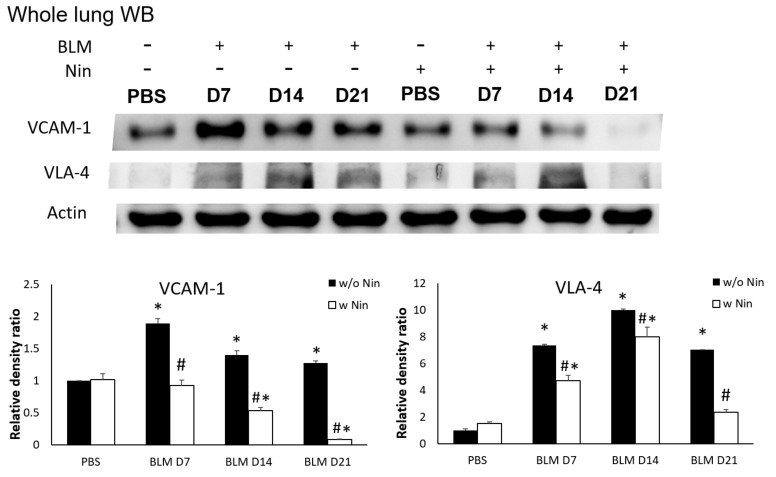
Nintedanib prevents neutrophil recruitment in mice with bleomycin-induced pulmonary fibrosis. Western blotting showed that both very late antigen 4 (VLA-4) and vascular cell adhesion molecule 1 (VCAM-1) levels were significantly higher seven days after bleomycin (BLM) injection. Nintedanib (Nin) treatment also ameliorated both the expression of VLA-4 and VCAM-1 seven days after bleomycin injection. * *p* < 0.05 compared to the PBS group; # *p* < 0.05 compared to the BLM w/o Nin group, *n* = 5 per group. PBS: phosphate-buffered saline, w/o: without, w/: with.

**Figure 6 ijms-21-04735-f006:**
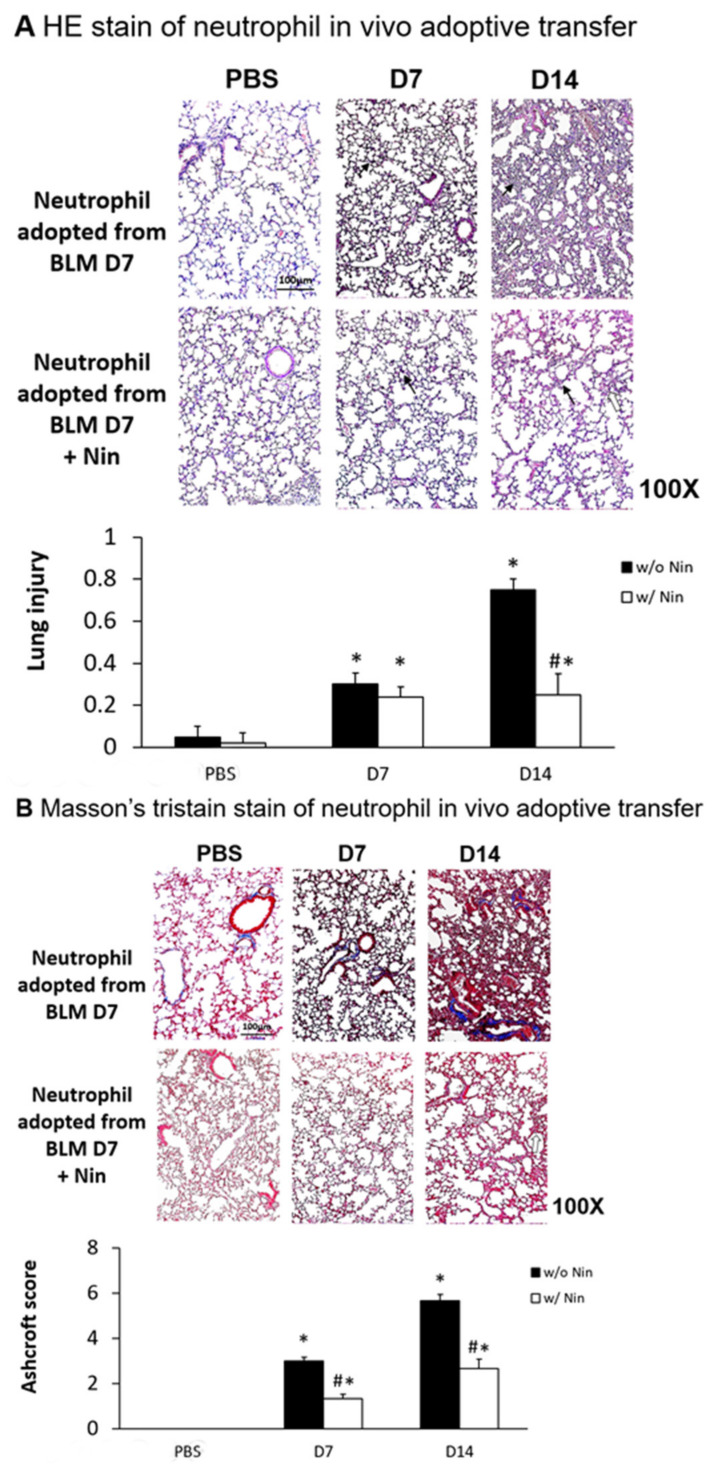
Nintedanib regulates lung inflammation and fibrosis in a mouse model of neutrophil adoptive transfer in vivo. Neutrophils (5 × 10^5^ cells) isolated from mice seven days after intratracheal injection with bleomycin (BLM) were intravenously administered to healthy mice. (**A**) Hematoxylin and eosin (HE) staining and lung injury scores demonstrated that lung inflammation (arrows: neutrophils infiltrations in lung tissue) and fibrosis (open arrows: alveolar fibrosis) were attenuated in mice receiving nintedanib (Nin) compared with the group receiving adoptive neutrophil transfer. (**B**) Mice that received nintedanib had significantly less pulmonary fibrosis (open arrows: alveolar fibrosis) than mice receiving adoptive neutrophil transfer according to Masson’s trichrome staining and the Ashcroft score. * *p* < 0.05 compared to the PBS group; # *p* < 0.05 compared to the BLM w/o Nin group, *n* = 5 per group. PBS: phosphate-buffered saline, w/o: without, w/: with.

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
