# Peer review of "Nintedanib Reduces Neutrophil Chemotaxis via Activating GRK2 in Bleomycin-Induced Pulmonary Fibrosis"

_ijms, 2020, doi:10.3390/ijms21134735_

Round 1

Reviewer 1 Report

Dear authors, you present here an interesting study regarding the effect of nintedanib in lung fibrosis. The paper is well written, still there are some aspects that I bring to your attention:

Lines 30, 36, 127, 131: „in vivo” should be written in italic

Line 33: expression, as well…

activity, in….

Line 36 : chemotaxis, to regulate….

Line 122 : Western blots

Line 220 : activity, but also exerted

Line 223 : ‘’in silica’’ should be in italic

I think that lines 239-246 belong to the Introduction part, rather than Discussion

Line 252: Secondly,…

I suggest using a reference for the anti-inflammatory effect, a well-known and used drug, in order to compare the effect of nintedanib.

I recommend that you develop the Conclusions part of the manuscript.

Author Response

Dear Reviewer 1

The manuscript has been revised in response to your comments.

Reviewer 1

Dear authors, you present here an interesting study regarding the effect of nintedanib in lung fibrosis. The paper is well written, still there are some aspects that I bring to your attention:

Lines 30, 36, 127, 131: „in vivo” should be written in italic

Answer: Thank you for your comment. We have revised the manuscript for correct writing. Please read Lines 31, 38, 139, and 143 in the revised manuscript.

Line 33: expression, as well…

activity, in….

Answer: Thank you for your comment. We have revised the manuscript for correct grammar use. Please read Lines 34 in the revised manuscript.

Line 36 : chemotaxis, to regulate….

Answer: Thank you for your comment. We have revised the manuscript for correct grammar use. Please read Line 39 in the revised manuscript.

Line 122 : Western blots

Answer: Thank you for your comment. We have revised the manuscript for correct word use. Please read Line 131 in the revised manuscript.

Line 220 : activity, but also exerted

Answer: Thank you for your comment. We have revised the manuscript for correct word use. Please read Line 242 in the revised manuscript.

Line 223 : ‘’in silica’’ should be in italic

Answer: Thank you for your comment. We have revised the manuscript for correct writing. Please read Line 253 in the revised manuscript.

I think that lines 239-246 belong to the Introduction part, rather than Discussion

Answer: Thank you for your comment. We have revised the manuscript in introduction and discussion section. Please read Line 70-77 in the introduction section and Lines 268-270 in discussion section in the revised manuscript.

Line 252: Secondly,…

Answer: Thank you for your comment. We have revised the manuscript for correct writing. Please read Line 281 in the revised manuscript.

I suggest using a reference for the anti-inflammatory effect, a well-known and used drug, in order to compare the effect of nintedanib.

Answer: Thank you for your comment. Steroid had been used in idiopathic pulmonary fibrosis, although without mortality benefit. We have added the comparison with nintedanib and steroid. Please read Lines 271-277 in the discussion section in the revised manuscript.

I recommend that you develop the Conclusions part of the manuscript.

Answer: Thank you for your comment. Nintedanib has dual effect on both regulating neutrophil chemotaxis and decreased activation of neutrophils. Please read Lines 38-42 and Lines 405-408 in the revised manuscript.

Reviewer 2 Report

The manuscript by Dr. Chen et al explores the effect of Nintedanib on pulmonary fibrosis from a mechanistic point of view, finding important confirmatory results on the reduction of fibrosis following the drug treatment. In addition, they found that Nintedanib seems to impair neutrophils recruitment in the lung by interfering with their chemotaxis. The results partially supports their thesis and add important knowledge in the field of idiopathic pulmonary fibrosis.

In my opinion, there are just few major and minor issues.

Major issues:

1) The outline of the experiments involving mice are not described well in material and methods section. For instance, you said that the mice received the intratracheal injection of bleomycin sulfate to induce the fibrosis. You also said that the control mice received an intratracheal injection of PBS. Did you use this “control” as baseline (so time 0) for the treatment? If so, the term control should be changed to baseline to avoid confusion. In fact, the real control group in these experiments is represented by the bleomycin induced fibrosis mice not treated with Nin. The results substantially do not change, it is just a matter of terminology to avoid confusion.

2) On Figure 2, for what I see, IL-1beta (panel A) seems really not increased when compared with baseline in Nin treated mice, but the opposite is true for the not treated group. On the contrary, MIP-2 seems increased in both treated and not treated mice when compared with baseline, but the treatment with Nin decreased the maximum levels reached at 14 days. How do you explain this different behavior in the two cytokines? In addition, by following the previous suggestion (see comment 1), in my opinion you should refer to baseline instead of control and to not treated with Nin (or other term you prefer) for the fibrosis group. In fact, both groups have bleomycin-induced lung fibrosis but in the treated one it is not evident due to the treatment.

3) Although the quality of the chosen representative blot for VLA-4 is questionable, the summary evidence that there is a decrease in the expression of VLA-4 following Nin treatment (compared with the w/o Nin), but the increase in its expression should be still significant when compared with the “control” (so the baseline, see previous comments). Do you think that this effect is sufficient to provide a decreased accumulation on neutrophils in the lungs (see also comment 4 and 5)?

4) Since you observed a decrease in CXCR2, you should check if Nin treatment is able to decrease IL-8 expression in the lungs. This would provide more insights about the mechanistic reasons on how Nin acts on neutrophils.

5) From the results of the manuscript, I see that Nin treatment cause a decrease in the accumulation of neutrophils in the lung, probably by interfering with their chemotaxis. I say probably because there is not a direct demonstration of an effect of Nin on isolated neutrophils to asses their chemotaxis in response to stimuli. In fact, in order to truly confirm this, the authors are encouraged to perform experiments on isolated neutrophils, although they have a hint from the upregulation of GRK2.

Collectively from the results obtained, it is in my opinion that the effect of Nin is dual. 1) it might be mainly ascribed to a decreased activation of endothelial cells as observed by the results from Figure 5 showing a decreased expression of VCAM-1 (and tentatively to a decreased production of chemokines directed towards neutrophils, e.g. IL-8). 2) At the same time, Nin could reduce the chemotaxis of neutrophils by decreasing the expression of CXCR2 and VLA-4, as well as increasing the expression of GRK2. However, without the direct experiments suggested above (on isolated neutrophils), it is quite difficult to draw a definite conclusion. I suggest to further discuss this possible dual action of Nin since it seems not adequately stressed right now in the discussion. In addition, if you are not able to perform the experiments on the chemotaxis of isolated neutrophils you should include it as a limitation of the study that may have prevented you to draw a definite conclusion.

Minor issues:

On line 48, at the beginning please change to “Patients affected by idiopathic pulmonary fibrosis (IPF), the most common form of ILD, have…”

Line 64, change “and sera in patients” with  “and sera of patients”.

On line 326, please write the concentration of Tween 20 used in the blocking buffer for western blot.

On lines 329-331, please write the code of the ECL used along with the code and dilution of the secondary antibody.

On line 337-338 please write the dilution of primary antibodies used for immunofluorescence.

On section 4.10, please write the dilution of antibodies used in the experiments.

On section 4.11, please write the dilution of antibodies used in the experiments.

On section 4.12, please write the catalog number and company of the kit used for cell sorting.

On pictures of IHC staining (Figure 1, 2 and 3), consider to put arrows to direct the reader towards the described events (e.g. for neutrophil infiltration, fibrosis and so on).

On section 2.2, lines 93-94 you said that you used IHC staining to detect changes in various cytokines. As far as I know, alfa-SMA and collagen-1 are not cytokines but can be referred as generic proteins. So please put the right term “proteins” instead of cytokines.

Consider to put the respective Figures right after the corresponding text in the results. For instance, you can put Figure 1 right after section 2.2, and so on.

In Figure 3, the letter B on the top left side of the histogram is somewhat confusing, since it may cause the reader to believe that this is the panel B. Instead, if I’m not wrong it is the graphical summary of the IF results above (from panel 3A). So, please remove the letter.

Author Response

Dear Reviewer 2

The manuscript has been revised in response to your comments.

Reviewer 2

Major issues:

1) The outline of the experiments involving mice are not described well in material and methods section. For instance, you said that the mice received the intratracheal injection of bleomycin sulfate to induce the fibrosis. You also said that the control mice received an intratracheal injection of PBS. Did you use this “control” as baseline (so time 0) for the treatment? If so, the term control should be changed to baseline to avoid confusion. In fact, the real control group in these experiments is represented by the bleomycin induced fibrosis mice not treated with Nin. The results substantially do not change, it is just a matter of terminology to avoid confusion.

Answer: Thank you for your comment. To avoid confusion, we replace the term of “control” to “PBS” as the mice received an intratracheal injection of 50μL PBS but no bleomycin. In designated experiments, mice received either nintedanib (suspended in 300 μL 0.5% hydroxyethyl-cellulose (HEC)) (referred to as w/ Nin-treated mice), or HEC 300μL only (referred to as w/o Nin-treated mice) via oral gavage. Therefore, the treatment-control group in this experiment is represented by the bleomycin induced fibrosis mice without nintedanib treatment but receiving oral HEC only (w/o Nin mice). Please read Lines 300-304 in 4.2 Animal model section, Figure 1A-D, 2A-B, 3A-C, 4A-B, 5, 6A-B and Figure legends (Lines 165-166, 177-179, 195-197, 208-209, 216-217, and 230-231) in the revised manuscript.

2) On Figure 2, for what I see, IL-1beta (panel A) seems really not increased when compared with baseline in Nin treated mice, but the opposite is true for the not treated group. On the contrary, MIP-2 seems increased in both treated and not treated mice when compared with baseline, but the treatment with Nin decreased the maximum levels reached at 14 days. How do you explain this different behavior in the two cytokines? In addition, by following the previous suggestion (see comment 1), in my opinion you should refer to baseline instead of control and to not treated with Nin (or other term you prefer) for the fibrosis group. In fact, both groups have bleomycin-induced lung fibrosis but in the treated one it is not evident due to the treatment.

Answer: Thank you for your comment. In addition to neutrophils, both IL-1 beta and MIP-2 were produced from various cell types and contributed to the fibrotic process. Nintedanib exerted different extent of inhibitory effect on theses cytokines. We have added the relevant discussion in the revised manuscript. Please read Lines 243-248 in the revised manuscript. As mentioned above (see comment 1), we replace the term of “control” to “PBS” as the mice received an intratracheal injection of 50μL PBS but no bleomycin. The treatment-control group in this experiment is represented by the bleomycin induced fibrosis mice without nintedanib treatment (w/o Nin mice). Please read Lines 300-304 in 4.2 Animal model section, Figure 1A-D, 2A-B, 3A-C, 4A-B, 5, 6A-B and Figure legends (Lines 165-166, 177-179, 195-197, 208-209, 216-217, and 230-231) in the revised manuscript.

3) Although the quality of the chosen representative blot for VLA-4 is questionable, the summary evidence that there is a decrease in the expression of VLA-4 following Nin treatment (compared with the w/o Nin), but the increase in its expression should be still significant when compared with the “control” (so the baseline, see previous comments). Do you think that this effect is sufficient to provide a decreased accumulation on neutrophils in the lungs (see also comment 4 and 5)?

Answer: Thank you for your comment. Nintedanib decreased accumulation of neutrophils in the lungs via two main mechanisms observed in our study. Firstly, nintedanib mediated a downregulation of expression of CXCR2 and chemotaxis in peripheral blood neutrophils in BLM mice. Secondly, nintedanib decreased both VLA-4 expression on neutrophils and VCAM-1 expression on endothelial cells to reduce neutrophils trans-endothelial migration. Taken together, these findings could explain the phenomenon of decreased neutrophils accumulation in the lungs. Please read Lines 268-270 in the revised manuscript.

4) Since you observed a decrease in CXCR2, you should check if Nin treatment is able to decrease IL-8 expression in the lungs. This would provide more insights about the mechanistic reasons on how Nin acts on neutrophils.

Answer: Thank you for your comment. Interleukin (IL)-8 has an important role in initiating inflammation in humans as it could attract immune cells such as neutrophils through their receptors CXCR1 and CXCR2. Though mice do not have the IL-8 gene, MIP-2 could be regarded as an IL-8 homologue in mice. MIP-2 has been found to contribute to a number of neutrophil-dependent disease in mice model. Please read Lines 248-251 in the revised manuscript.

5) From the results of the manuscript, I see that Nin treatment cause a decrease in the accumulation of neutrophils in the lung, probably by interfering with their chemotaxis. I say probably because there is not a direct demonstration of an effect of Nin on isolated neutrophils to asses their chemotaxis in response to stimuli. In fact, in order to truly confirm this, the authors are encouraged to perform experiments on isolated neutrophils, although they have a hint from the upregulation of GRK2.

Answer: Thank you for your comment. It is the limitation of this study. We have added it in the limitation section. Please read Lines 285-287 in the revised manuscript.

Collectively from the results obtained, it is in my opinion that the effect of Nin is dual. 1) it might be mainly ascribed to a decreased activation of endothelial cells as observed by the results from Figure 5 showing a decreased expression of VCAM-1 (and tentatively to a decreased production of chemokines directed towards neutrophils, e.g. IL-8). 2) At the same time, Nin could reduce the chemotaxis of neutrophils by decreasing the expression of CXCR2 and VLA-4, as well as increasing the expression of GRK2. However, without the direct experiments suggested above (on isolated neutrophils), it is quite difficult to draw a definite conclusion. I suggest to further discuss this possible dual action of Nin since it seems not adequately stressed right now in the discussion. In addition, if you are not able to perform the experiments on the chemotaxis of isolated neutrophils you should include it as a limitation of the study that may have prevented you to draw a definite conclusion.

Answer: Thank you for your comment and instruction. We have added the limitation section about in vitro experiment. Please read Lines 285-287 in the revised manuscript. We also re-written our conclusion section. Please read Lines 38-42 and Lines 405-408 in the revised manuscript.

Minor issues:

On line 48, at the beginning please change to “Patients affected by idiopathic pulmonary fibrosis (IPF), the most common form of ILD, have…”

Answer: Thank you for your comment. We have revised the manuscript for more accurate phrases. Please read Line 51 in the revised manuscript.

Line 64, change “and sera in patients” with  “and sera of patients”.

Answer: Thank you for your comment. We have revised the manuscript for correct writing. Please read Line 68 in the revised manuscript.

On line 326, please write the concentration of Tween 20 used in the blocking buffer for western blot.

Answer: Thank you for your comment. The concentration of Tween 20 is 1%. Please read Lines 357 in the revised manuscript.

On lines 329-331, please write the code of the ECL used along with the code and dilution of the secondary antibody.

Answer: Thank you for your comment. The code of the ECL was 32106 and the dilution of the secondary antibody was 1:1000. Please read Lines 361-363 in the revised manuscript.

On line 337-338 please write the dilution of primary antibodies used for immunofluorescence.

Answer: Thank you for your comment. The dilution of primary antibodies used for immunofluorescence was 1:100 for Ly6G and 1:100 for CXCR2. Please read Lines 368-369 in the revised manuscript.

On section 4.10, please write the dilution of antibodies used in the experiments.

Answer: Thank you for your comment. To examine surface CXCR2 expression, mouse whole blood cells with various treatments were stained with equal FITC-labeled anti-Ly6G and Alexa647-labeled anti-CXCR2 antibodies (0.125 μg per 106 cells, BioLegend, San Diego, CA). Besides, the dilution of GRK2 antibody was 1:50. Please read Lines 377-379 and 382 in the revised manuscript.

On section 4.11, please write the dilution of antibodies used in the experiments.

Answer: Thank you for your comment. The dilution of GRK2 antibody is 1:100 and the dilution of the secondary antibody is 1:400. Please read Lines 388-389 in the revised manuscript.

On section 4.12, please write the catalog number and company of the kit used for cell sorting.

Answer: Thank you for your comment. The kit used for cell sorting is 130-092-332, Miltenyi Biotec, Germany, Bergisch Gladbach. Please read Lines 395-396 in the revised manuscript.

On pictures of IHC staining (Figure 1, 2 and 3), consider to put arrows to direct the reader towards the described events (e.g. for neutrophil infiltration, fibrosis and so on).

Answer: Thank you for your comment. We have put arrows, open arrows and arrowheads for better describing the events mentioned on HE stain. Please read Figure 1A, 1B, 6A and 6B and Figure legend (Lines 155-157, 159, 225-226, 228-229) in the revised manuscript.

On section 2.2, lines 93-94 you said that you used IHC staining to detect changes in various cytokines. As far as I know, alfa-SMA and collagen-1 are not cytokines but can be referred as generic proteins. So please put the right term “proteins” instead of cytokines.

Answer: Thank you for your comment. We have revised the manuscript for correct wording. Please read Line 106 in the revised manuscript.

Consider to put the respective Figures right after the corresponding text in the results. For instance, you can put Figure 1 right after section 2.2, and so on.

Answer: Thank you for your comment. The current format is provided for peer review only and is suggested by the journal. The presentation of figures will be further formatted right after each section if accepted for publication.

In Figure 3, the letter B on the top left side of the histogram is somewhat confusing, since it may cause the reader to believe that this is the panel B. Instead, if I’m not wrong it is the graphical summary of the IF results above (from panel 3A). So, please remove the letter.

Answer: Thank you for your comment. We have modified our Figure 3. Please read Figure 3 in the revised manuscript.

Round 2

Reviewer 2 Report

I feel that all my concerns have been addressed in this round of review.